# Upregulation of large myelin protein zero leads to Charcot–Marie–Tooth disease-like neuropathy in mice

Yoshinori Otani [1], Nobuhiko Ohno [2,3], Jingjing Cui [1], Yoshihide Yamaguchi [1✉] & Hiroko Baba [1]

Charcot–Marie–Tooth (CMT) disease is a hereditary neuropathy mainly caused by gene mutation of peripheral myelin proteins including myelin protein zero (P0, MPZ). Large myelin protein zero (L-MPZ) is an isoform of P0 that contains an extended polypeptide synthesized by translational readthrough at the C-terminus in tetrapods, including humans. The physiological role of L-MPZ and consequences of an altered L-MPZ/P0 ratio in peripheral myelin are not known. To clarify this, we used genome editing to generate a mouse line (L-MPZ mice) that produced L-MPZ instead of P0. Motor tests and electrophysiological, immunohistological, and electron microscopy analyses show that homozygous L-MPZ mice exhibit CMT-like phenotypes including thin and/or loose myelin, increased small-caliber axons, and disorganized axo–glial interactions. Heterozygous mice show a milder phenotype. These results highlight the importance of an appropriate L-MPZ/P0 ratio and show that aberrant readthrough of a myelin protein causes neuropathy.

[1] Department of Molecular Neurobiology, Tokyo University of Pharmacy and Life Sciences, Hachioji, Japan. [2] Department of Anatomy, Division of Histology and Cell Biology, School of Medicine, Jichi Medical University, Shimotsuke, Japan. [3] Division of Neurobiology and Bioinformatics, National Institute for Physiological Sciences, Okazaki, Japan. ✉email: yoshiy@toyaku.ac.jp

Myelin acts as an electrical insulator to promote saltatory conduction of action potentials[1]. To perform this function, myelin-forming cells are involved in the formation and maintenance of functional axonal domains including the node of Ranvier, paranode, juxtaparanode, and internode. In the peripheral nervous system (PNS), myelin is produced by Schwann cells. To form or maintain myelin, unique cytoplasmic channels are present in PNS myelinated fibers, including Schmidt–Lanterman incisures (SLI) in myelin and Cajal bands in the outermost regions of myelinated Schwann cells[1]. Disorganization of these cytoplasmic channels is frequently seen in knockout mouse lines expressing neuropathy-like phenotypes[2–5].

Myelin protein zero (P0, also called MPZ) is a major peripheral myelin protein that acts as a homophilic adhesion molecule, and is crucial for compact myelin formation in the PNS[6]. Adhesion activity is regulated by protein kinase C (PKC)-dependent phosphorylation in the cytoplasmic region of P0[7]. The *MPZ* gene is one of the genes responsible for Charcot–Marie–Tooth (CMT) disease, which is also known as hereditary motor and sensory neuropathy (HMSN). Traditionally, CMT disease caused by *MPZ* mutations is categorized into three types based on electrophysiological characteristics, specifically: demyelinating, axonal, and intermediate[8] (https://www.ncbi.nlm.nih.gov/books/NBK1358/). Many P0 mutations cause various types of HMSN, including CMT1B[9]. Additionally, heterozygote and homozygote P0 knockout mice exhibit CMT1B (demyelinating), or the more severe, Dejerine–Sottas syndrome-like phenotype, respectively[10–12]. Therefore, P0 is an essential molecule for myelin formation and maintenance in the PNS.

Large myelin protein zero (L-MPZ) is a recently identified translational readthrough isoform of P0, which has an additional 63-amino acid sequence at the C-terminus of P0[13]. Moreover, this L-MPZ-specific domain contains a putative PKC phosphorylation site with high antigenicity. L-MPZ is expressed during myelinogenesis, is localized in PNS myelin, and accumulates at cell–cell adhesion sites in transfected cells, suggesting that L-MPZ is involved in the formation and/or maintenance of myelin in the PNS. Because the sequences around the canonical stop codon and L-MPZ-specific domain of *MPZ* genes (*mpz*, *Mpz*) are highly conserved from frog to human in the vertebrate phylogeny, and L-MPZ is present within the PNS of these species[14], the appearance of L-MPZ may be important for myelin during the evolution of tetrapod limbs.

Translational readthrough is a key mechanism in translational regulation, and is thought to expand the coding potential of restricted genomes in viruses[15], yeasts[16], and *Drosophila*[17]. To our knowledge, L-MPZ is the first reported common mammalian protein synthesized by the stop codon readthrough mechanism in the physiological state[13]. Following identification of L-MPZ, two additional, widely expressed mammalian readthrough isoforms were reported, namely extended isoforms of vascular endothelial growth factor A (VEGF-Ax)[18] and aquaporin 4 (AQP4ex)[19]. Thus, translational readthrough may be a common physiological mechanism for translational regulation in mammals including humans[20]. Interestingly, VEGF-Ax has anti-angiogenic or weaker angiogenic activity compared with canonical VEGF[18,21], while AQP4ex displays a crucial role for the anchoring of canonical AQP4 at the astrocyte end-feet and potential regulation of water permeability[19,22]. Together, these findings suggest that quantitative changes of translational readthrough isoforms affect the function of canonical molecules. However, because both P0 and L-MPZ are produced from the same P0 mRNA and are both localized in myelin, it is difficult to determine the functional significance of L-MPZ itself, and the influence of the L-MPZ/P0 ratio on myelin in vivo is still uncertain.

Accordingly, in this study, we sought to clarify the functional significance of changes in amount of the readthrough isoform, L-MPZ, in the PNS. Hence, we used a genome editing system to generate mice (L-MPZ mice) with mutation of the canonical stop codon in the *Mpz* gene. As a result, these mice synthesize L-MPZ but not P0. Increased L-MPZ levels in heterozygous (Het) and homozygous (Hom) L-MPZ mice cause severe morphological and functional changes of myelin and axons that are similar to CMT pathologies.

## Results

**Generation of L-MPZ mice**. To distinguish the function of L-MPZ from P0, we used the clustered regularly interspaced short palindromic repeats–CRISPR associated protein 9 (CRISPR–Cas9) genome editing system to generate mice that synthesize only L-MPZ, and not P0, by replacement of the canonical stop codon (TAG) with an alanine codon (GCG) (Supplementary Fig. 1a). Two founder L-MPZ mice with germline transmission were backcrossed with C57BL/6J mice, and individual mouse lines were independently maintained. Genotypes of the L-MPZ mice were determined by allele-specific polymerase chain reaction (PCR), as described in the Methods section (Supplementary Fig. 1b). To determine whether L-MPZ mice synthesized only L-MPZ, we examined P0 and L-MPZ levels in the sciatic nerve by sodium dodecyl sulfate–polyacrylamide gel electrophoresis (SDS–PAGE) using Coomassie Brilliant Blue staining (Supplementary Fig. 1c) and western blotting (Supplementary Fig. 1d; uncropped blots, Supplementary Fig. 10a–c). Normally, L-MPZ levels are approximately 5–10% of P0 in wild–type (WT) mice. However, L-MPZ Hom mice showed complete replacement of P0 (~30 kDa) with L-MPZ (~36 kDa) in sciatic nerve homogenates, while L-MPZ Het mice produced both L-MPZ and P0 proteins at similar levels (Supplementary Fig. 1c, d). These results indicate that the L-MPZ/P0 ratio is increased in the following order: Hom > Het > WT mice.

**Motor dysfunctions in L-MPZ mice**. L-MPZ mice were born in the expected Mendelian ratios and were indistinguishable from WT littermates during development, indicating that replacement of P0 with L-MPZ does not affect embryonic and early postnatal development. Previously, P0 knockout mice were shown to exhibit hindlimb clasping by tail suspension[10]. Therefore, to examine motor function in 10-week-old L-MPZ mice, we used the modified tail suspension test (as described in the Methods section). Angles formed between bilateral hindlimbs (leg angle: Fig. 1a, WT 1–2 min) were measured at two time points: 1–2 min (early period) and 4–5 min (later period) after starting the test (Fig. 1a, b; Supplementary Table 1; Supplementary Movies 1–6). When suspended by their tails for 6 min, most WT mice kept their hindlimbs wide open as well as their toes. Similarly, L-MPZ Het mice tended to keep their hindlimbs and toes open at the early period, but gradually closed their hindlimbs, with smaller angles than WT mice at later periods. In contrast, L-MPZ Hom mice kept their hindlimbs closed throughout the test. Next, we used the rotarod test to record latency to fall from a rotating rod. This parameter was independently recorded three times, and found to be shorter compared with WT mice (Fig. 1c; Supplementary Table 1). Altogether, these results indicate that both L-MPZ Hom and Het mice exhibit motor disturbances, although Het mice show a milder phenotype than Hom mice. Motor nerve conduction velocities (NCVs) were then measured in the plantar muscle of L-MPZ Het and Hom mice, and found to be slower compared with control WT mice (Fig. 1d, e; Supplementary Table 1). Further, amplitudes of compound muscle action potentials (CMAPs) were reduced in both types of L-MPZ mice,

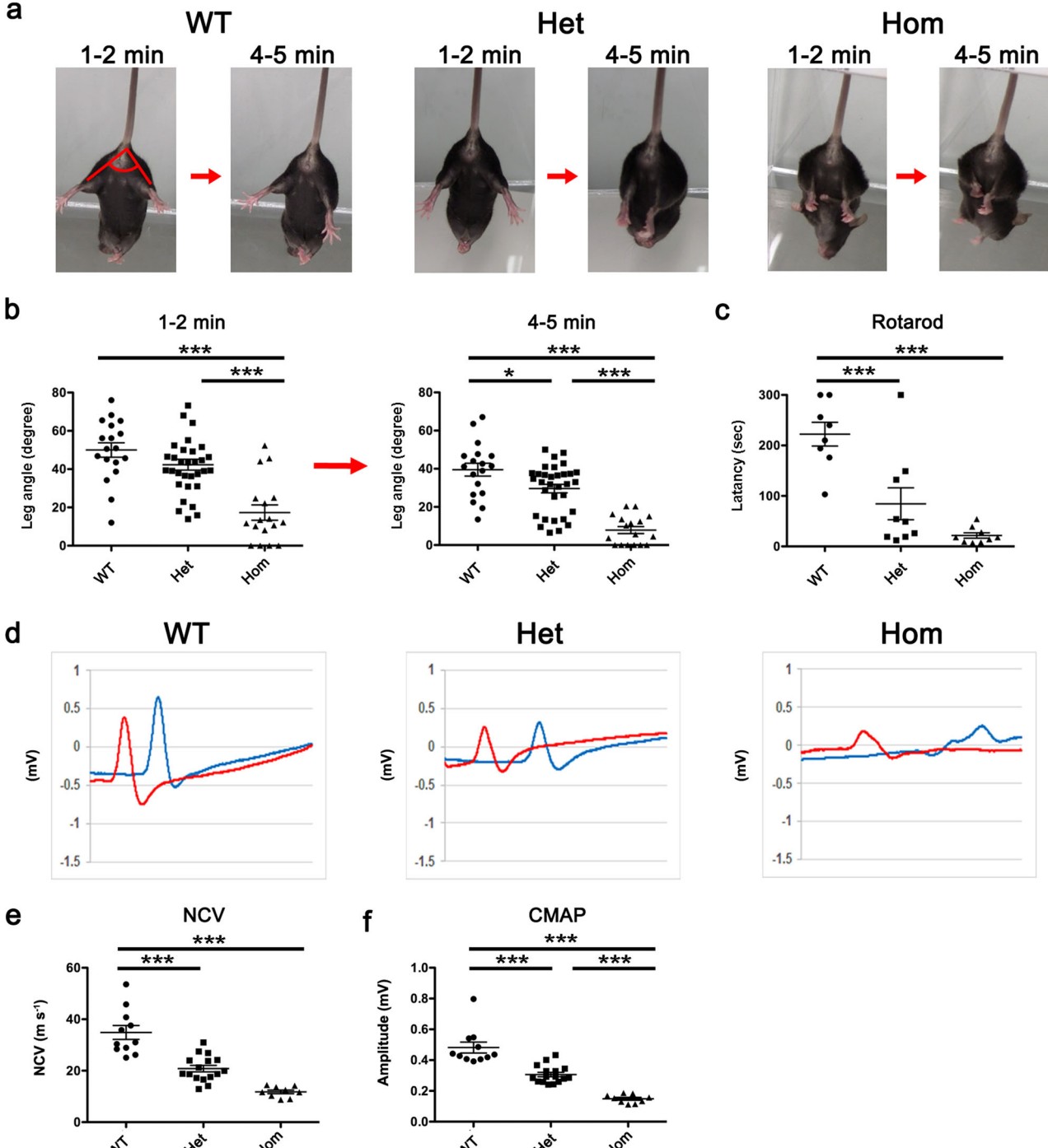

**Fig. 1 Motor disturbance and motor nerve dysfunction in L-MPZ mice. a** Representative postures for each mouse genotype at early (1–2 min) and later (4–5 min) time points. To investigate motor function of L-MPZ mice, the tail suspension test was performed for 6 min using 10-week-old L-MPZ wild-type (WT), heterozygous (Het), and homozygous (Hom) mice. Angles formed between bilateral hindlimbs are indicated as red lines in the left image (1–2 min, WT). **b** Comparison of motor function by bilateral hindlimb angles indicated in **a** at two time points using video-recorded pictures. **c** Comparison of motor function examined by the rotarod test. The rotation speed of the rod was 20 rpm over 5 min in each trial. Latency to the time when the mouse fell from the rod was measured. **d** Representative waveform sets of compound muscle action potentials (CMAPs) obtained from individual genotypes. Motor nerve conduction study was performed in L-MPZ mice. The left or right tibial branch of the sciatic nerve was electrically stimulated at the ankle (**d**, red line) or sciatic notch (**d**, blue line). **e**, **f** Comparison of average left and right sciatic nerve conduction velocities (NCV; **e**) and CMAPs (**f**) obtained from individual genotypes. *$p < 0.05$; ***$p < 0.001$ by ANOVA with post-hoc Tukey's test. Data are presented as mean ± SE of experiments. N in WT = 16, Het = 24, Hom = 14 (**b**); WT = 8, Het = 9, Hom = 9 (**c**); and WT = 16, Het = 24, and Hom = 9 (**d–f**).

with abnormal temporal dispersion observed in Hom mice (Fig. 1d and f; Supplementary Table 1).

Because L-MPZ mice show motor disturbances, morphological changes in muscle fibers were examined in hematoxylin and eosin-stained muscle tissue. Some muscle fibers displayed central nuclei indicating as regeneration in Hom mice, but overall grouped muscle fiber atrophy (group atrophy) was more prominent in Hom than Het mice. These results indicate that neurogenic muscular atrophy has developed in L-MPZ mice (Supplementary Fig. 2).

**Disruption of PNS myelin structure in L-MPZ mice**. To determine if myelin sheaths were affected by replacement of P0 with L-MPZ, we examined myelin structure by immuno-fluorescence. At 10 weeks of age, sciatic nerves of L-MPZ mice demonstrated increased L-MPZ signal intensity. However, Hom mice showed a marked decrease of myelin basic protein (MBP)-positive fluorescence signal (Fig. 2a). Immunostaining of single teased fibers from L-MPZ Hom mice often showed axons with fragmented L-MPZ-positive or MBP-positive signals in both large and small sized axons (Fig. 2b, c), indicating either presence of extensive demyelination or fragile myelin that is easily broken in the L-MPZ Hom PNS.

One of the characteristic features of PNS myelin is the presence of cytoplasmic channels, such as SLIs or Cajal bands. SLIs are cytoplasm-containing funnel-like structures found in internodes of PNS myelin[1]. Their function is not fully understood, but as connexin 32-containing gap junctions are localized in SLIs, they may be involved in the transport of small molecules between Schwann cell bodies and the innermost myelin layer[23]. Thus, we investigated SLI structure in L-MPZ mice using E-cadherin as a marker of SLI containing adherens junctions[24]. Although both L-MPZ Hom and Het mice displayed decrease of E-cadherin-positive funnel-like staining, this staining pattern was virtually absent in Hom mice (Fig. 2d).

Cajal bands that run longitudinally on the surface of myelin internodes can be clearly distinguished from non-channel areas of the myelin surface by antibodies against periaxin or dystrophin-related protein 2 (DRP2)[25–27]. Loss of these proteins causes disorganization of Cajal bands and disturbs transport of MBP mRNA to bilateral paranodal regions[26]. To examine Cajal band structure, we immunostained teased sciatic nerve fibers with anti-DRP2 antibody (Fig. 2e). In WT mice, DRP2-negative channels were clearly demarcated by DRP2-positive cobblestone-like non-channel areas. In contrast, in L-MPZ mice, these structures were severely affected, especially in Hom mice. Using captured images, $f$-ratios that indicate the ratio of Cajal band area to DRP2-positive area were calculated (Fig. 2f). In normal mature sciatic nerves, the $f$-ratio is approximately 1[4]. While in both L-MPZ mouse genotypes, the $f$-ratio was increased (Fig. 2f; Supplementary Table 1), suggesting that the outer surface area of myelin covered by Schwann cell cytoplasmic channels was abnormally increased. Increased $f$-ratios have also been reported in mice deficient in either periaxin or laminin alpha-2 and dystroglycan[3,4]. Therefore, we investigated laminin alpha-2 and beta-dystroglycan levels in L-MPZ Hom mice by western blotting (Supplementary Fig. 3; uncropped blots, Supplementary Fig. 10d–f). We found decreased laminin alpha-2 and beta-dystroglycan (43 kDa [$DG_{43}$] and 31 kDa [$DG_{31}$]) in L-MPZ Hom mice. Taken together, both cytoplasmic channels of myelin internodes are disorganized in L-MPZ mice.

Next, we examined the detailed myelin structure of sciatic nerves using semithin and ultrathin sections by light microscopy and electron microscopy (EM), respectively. In semithin sections, compact myelin sheaths were thinner in L-MPZ mice compared

with WT mice (Fig. 3a, b, arrows), with small myelinated axons more prominent in L-MPZ mice (Fig. 3b, arrowheads). Meanwhile, EM analysis clearly showed similar sized axons that were covered by either thinner myelin sheaths (Fig. 3d, arrows) or Schwann cell bodies without myelin (Fig. 3d, black arrowhead) in L-MPZ Hom mice compared with WT mice (Fig. 3c). Furthermore, the surface of some myelinated axons was entirely covered by Schwann cell cytoplasm (Fig. 3d, white arrowheads), which is consistent with our results showing disappearance of DRP2-positive cobblestone-like structures and an increase in the $f$-ratio in L-MPZ mice (Fig. 2f). The number of axons individually covered by Schwann cells but lacking compact myelin was increased in L-MPZ mice (Fig. 3e and Supplementary Table 2). The $G$-ratio represents the ratio between axonal diameter ($D_{Ax}$) and total fiber diameter ($D_F$). In large-caliber axons, the $G$-ratio was particularly large (Fig. 3f, g and Supplementary Table 2) in L-MPZ Hom mice compared with WT mice. Additionally, the axonal diameters were decreased in Hom mice (Fig. 3h and Supplementary Table 2). These results indicate that both myelin thickness and axonal size are affected. Similar myelin and axonal abnormalities were also found in L-MPZ Het mice, although they were milder than those in Hom mice (Supplementary Fig. 4c, d).

In each layer of compact myelin, inner leaflets of apposed plasma membranes merged together to form one thick line called a major dense line (MDL) (Fig. 4a, dark lines in the insertion), while one intraperiod line (IPL) (Fig. 4a, light lines in the insertion) appeared between the outer leaflets of apposed membranes. Our EM study showed constant periodicity in WT myelin (Fig. 4a, double-headed arrows in the insertion). However, in L-MPZ Hom mice, MDLs failed to merge (Fig. 4b, dark lines in the insertion) and myelin outfoldings were also found (Fig. 4b, arrows). In some myelinated fibers, myelin showed a loose structure (Fig. 4c), with some larger spaces containing patchy electron-dense material inside unmerged MDLs (Fig. 4d, e, black arrowheads). Spiral membranes of normal compact myelin have a periodicity of approximately 13–14 nm when fixed by aldehydes and embedded in epoxy resin[25]. Periodicity was larger in L-MPZ Hom mice (Fig. 4f and Supplementary Table 2). In semithin sections, focal myelin swellings called tomacula were detected in L-MPZ Hom and Het mice (red and blue arrowheads in Supplementary Fig. 4a, b).

**Disorganization of nodal structures in L-MPZ mice**. The molecular organization at the node of Ranvier, paranode, and juxtaparanode were examined by immunofluorescence analysis using specific markers. Molecules including a glial 155 kDa isoform of neurofascin (NF155) and an axonal complex of contactin and contactin-associated protein 1 (Caspr/paranodin) are involved in paranodal axoglial junctions[28]. These paranodal junctions act as a diffusion barrier of juxtaparanodal proteins including voltage-gated $K^+$ channels such as Kv1.2[29–31]. Triple staining of WT teased fibers using antibodies against neurofascin, Caspr, and Kv1.2 showed a pair of Caspr-positive paranodal regions flanked by Kv1.2-positive juxtaparanodes. In contrast, various abnormal staining patterns were detected in L-MPZ Hom and Het mice (Fig. 5a), including shorter and irregular Caspr-positive staining with unstained gaps (white asterisk), longer Kv1.2-positive areas with abnormal gaps (white circle), and absence of Kv1.2 clusters (white #). Quantitative analysis indicates the shorter Caspr-positive paranodal and longer Kv1.2-positive juxtaparanodal lengths in L-MPZ mice compared with WT mice (paranodal and juxtaparanodal length graphs in Fig. 5d, e and Supplementary Table 1). Sodium channels are enriched at the nodal axolemma, and the 186 kDa isoform of neurofascin (NF186) is one of the key players in nodal assembly and

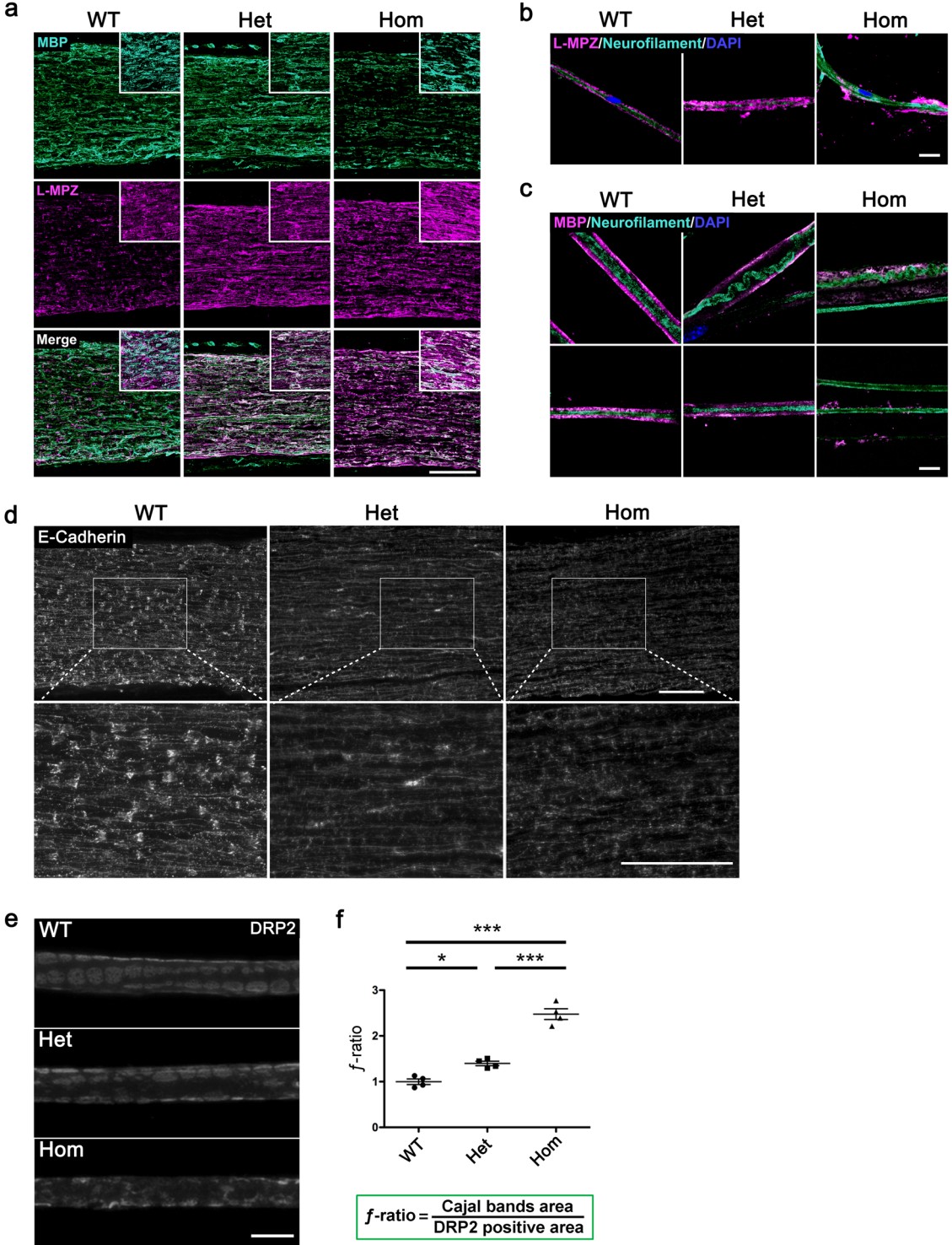

**Fig. 2 Abnormal myelin structure in L-MPZ mice by immunohistological analyses. a** Double immunostaining patterns of sciatic nerve sections from 10-week-old L-MPZ wild-type (WT), heterozygote (Het), and homozygote (Hom) mice using anti-MBP (turquoise) and L-MPZ (magenta) antibodies. **b**, **c** Triple immunostaining patterns of teased sciatic nerve fibers in L-MPZ mice. Teased fibers were immunostained using anti-neurofilament (turquoise) and L-MPZ (magenta; **b**), or MBP (magenta; representative two images, **c**) with DAPI (blue). **d** Visualization of Schmidt–Lanterman incisures (SLIs). Immunostaining of sciatic nerve sections in L-MPZ mice was performed using anti-E-cadherin antibody (white). The white square is enlarged. **e** Cajal bands visualized as unstained areas by anti-DRP2 antibody staining (white). DRP2-positive flanking regions are shown as a cobblestone appearance. **f** For quantification, the $f$-ratio (which indicates the ratio of Cajal band area [cytoplasm-containing channels] to DRP2 positive area [non-channel myelin surface]) was measured in teased sciatic nerve fibers. Bars, 100 μm (**a**, **d**) or 10 μm (**b**, **c**, **e**). *$p < 0.05$; ***$p < 0.001$ by one-way ANOVA with post-hoc Tukey's test. Data are presented as mean ± SE of experiments. $N$ in WT = 4, Het = 4, Hom = 4 (**f**). Data in **f** are presented as mean ± SE of four experiments.

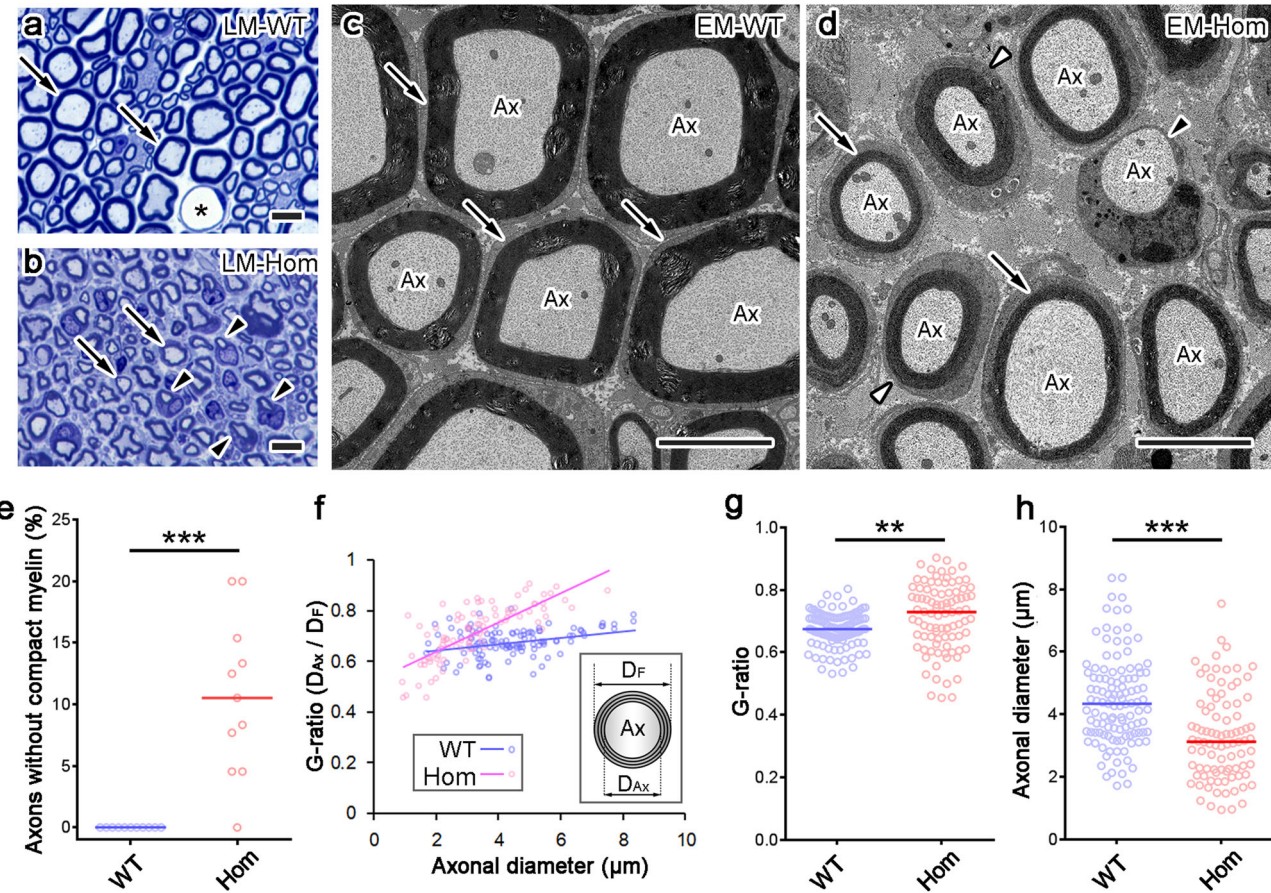

**Fig. 3 Increase of thin myelin, small axons, and axons without myelin in L-MPZ mice. a, b** Representative toluidine blue staining of 500 nm-thick sections of sciatic nerves in 8-week-old wild-type (WT) and homozygous (Hom) mice. Asterisk in **a** indicates the lumen of blood vessels. In light micrographs (LM), compared with WT (**a**, arrows), compact myelin appeared thinner (**b**, arrows), and smaller axons with myelin ensheathment (**b**, arrowheads) were prominent in Hom mice. **c, d** In electron micrographs (EM), axons (Ax) had thick compact myelin sheath in WT (**c**, arrows), while axons had thin myelin sheath (**d**, arrows) or lacked compact myelin (**d**, black arrowhead) in Hom mice. Cajal bands were not clearly formed around some axons in Hom mice (**d**, white arrowheads). **e** The number of axons that were individually ensheathed by Schwann cells but lacked compact myelin (**d**, black arrowhead) increased in Hom mice. **f-h** The $G$-ratio (obtained by dividing axonal diameter [$D_{Ax}$] with fiber diameter [$D_F$]) was larger (**f, g**) and axonal diameter smaller (**h**) in Hom mice. Bars, 10 μm (**a, b**) or 5 μm (**c, d**). **$p < 0.01$; ***$p < 0.001$ by Mann–Whitney $U$-test. $N$ (photo areas prepared from each of 4 mice) in WT = 11, Hom = 11 (**e**). $N$ (axons in photo areas) in WT = 115, Ho = 91 (**f-h**).

maintenance[32]. In Schwann cell microvilli, ERM (ezrin/radixin/moesin) family proteins and dystroglycan are involved in correct sodium channel clustering at the nodal axon and normal NCV[33–35]. Sodium channel clusters were present but nodal lengths became longer in L-MPZ mice compared with WT mice (Fig. 5b, and nodal length graphs in Fig. 5c; Supplementary Table 1). Further, moesin-positive signal intensity decreased, especially in L-MPZ Hom mice (Fig. 5b), suggesting that microvilli were severely disrupted. Furthermore, internodal lengths were shorter between two neighboring nodes distinguished by paranodal markers in L-MPZ mice compared with WT mice (Supplementary Fig. 5; Supplementary Table 1). The quantitative analysis of the number of abnormal node–paranode–juxtaparanode structures in sciatic nerve sections of L-MPZ mice revealed that molecular organization of the node of Ranvier and the surrounding area were clearly affected in L-MPZ compared with WT mice (Fig. 5f and Supplementary Table 3). Thus, replacement of P0 with L-MPZ causes disorganization of functional axonal domains and Schwann cell microvilli.

In EM, paranodal axo–glial junctions are characterized by intercellular transverse bands, which are regularly arrayed densities between the axolemma and lateral loops of myelin[36]. To examine the structural abnormalities causing molecular

changes, we performed serial block-face scanning electron microscopy (SBF-SEM) and transmission electron micrograph (TEM) examinations using longitudinal sciatic nerve sections from L-MPZ mice. In SBF-SEM analyses, serial images of a single myelinated axon in sciatic nerves of L-MPZ Hom mice (Supplementary Fig. 6a–f) identified three groups of paranodal loops intermittently visible near the nodal region towards the internode (Supplementary Fig. 6d–f, arrowheads). Compact myelin sheaths of different thickness flanked these groups of ectopic paranodal loops (Supplementary Fig. 6f, compare the thickness between the pairs of arrows). In TEM, some ectopic paranodal loops with or without electron-dense transverse bands were observed in L-MPZ mice (Supplementary Fig. 6g–k). These paranodal changes correspond appropriately to the short and occasionally interrupted staining patterns of NF155 and Caspr (Fig. 5a). Thus, continuous appearance of paranodal loops and the junctional structure between these loops and apposed axolemma are disturbed in sciatic nerves of L-MPZ mice.

**Increase of ER stress in sciatic nerves of L-MPZ mice.** Endoplasmic reticulum (ER) stress contributes to the pathogenesis of neuropathy including CMT[37]. Because L-MPZ has larger cytoplasmic region, complete replacement of P0 with L-MPZ may

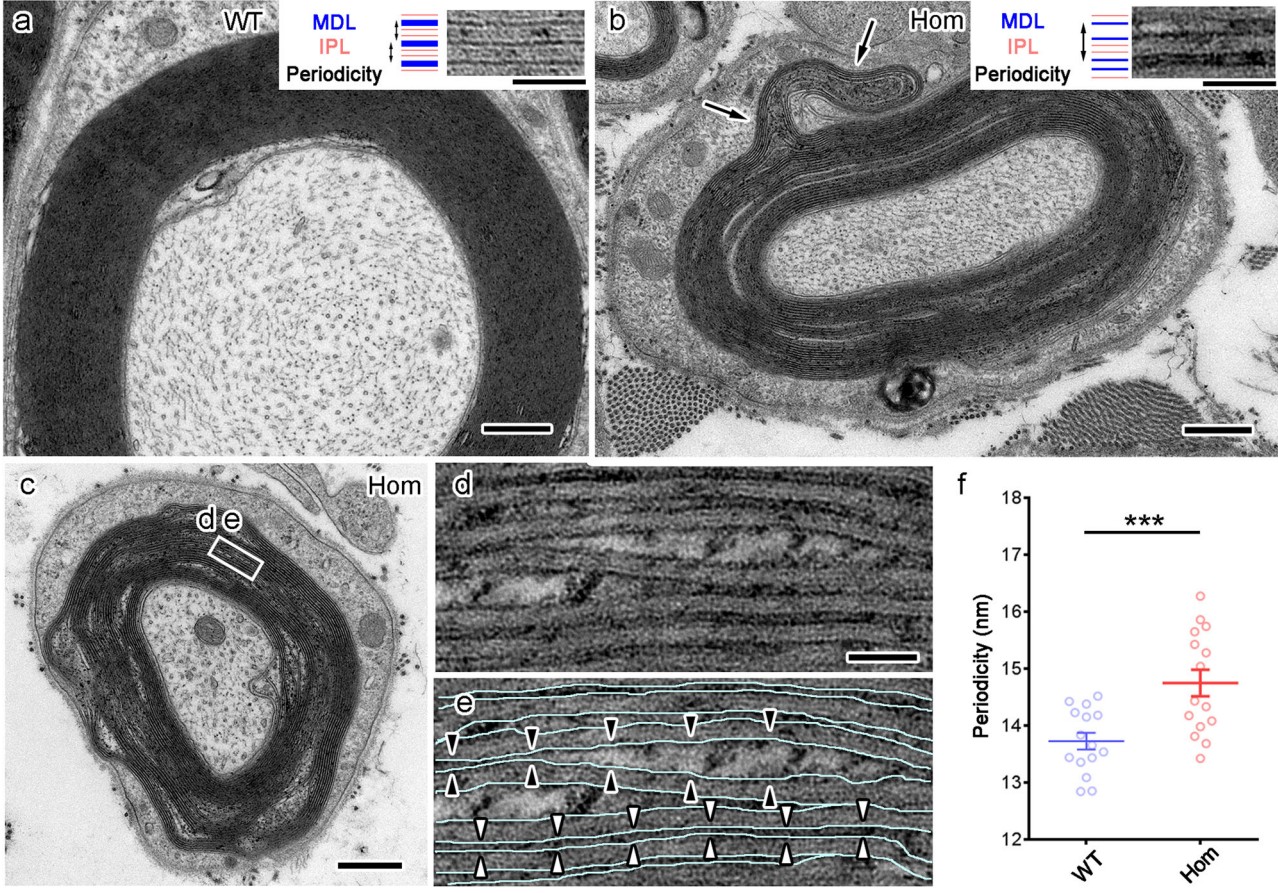

**Fig. 4 Ultrastructural abnormalities of compact myelin in L-MPZ mice.** Failed merging of major dense lines (MDL), increased periodicity of compact myelin, and outfoldings in 8-week-old L-MPZ homozygous (Hom) mice. **a** In the myelinated axon of a wild-type (WT) mouse, the MDL merged to form one thick dark line (MDL, blue lines in insertion) that was flanked by light intraperiod lines (IPL, space between red lines in insertion). Constant periodicity was observed (double headed arrows in insertion). **b** In a Hom mouse, the MDL failed to merge (blue separated lines in insertion) and periodicity increased (double-headed arrow in insertion). Myelin outfolding also formed (arrows). **c–e** In another myelinated fiber, MDL (**e**, light blue lines) did not merge (**e**, white arrowheads) and some larger spaces with patchy electron-dense materials were observed inside the unmerged major dense lines (**e**, black arrowheads). **f** Periodicity was larger in Hom. Bars, 500 nm (**a–c**) or 50 nm (**d, e**). ***$p < 0.001$ by Student's $t$-test. Data are presented as mean ± SE of experiments. $N = 15$ axons (**f**).

induce ER stress, which could cause myelin damage. Consequently, to examine ER stress levels, we performed western blotting and immunohistochemical analyses of sciatic nerves using anti-GRP78 antibody. As expected, GRP78 protein levels were markedly increased in L-MPZ Hom mice (Fig. 6a, b; uncropped blot, Supplementary Fig. 10g). Furthermore, increased GRP78 immunoreactivity was observed in sciatic nerve sections of L-MPZ Het mice compared with WT mice, in which no apparent GRP78-staining was observed (Supplementary Fig. 7). As another evidence of increase of ER stress in L-MPZ Hom mouse, anti-KDEL (additional ER marker) signals were increased in immunofluorescence staining of sciatic nerve sections using anti-KDEL antibody (Supplementary Fig. 8a, b). In addition, the swollen ER is usually observed in the ER stress-induced cells[38]. As a demonstration of such ER stress in the electron microscopic images, Schwann cells of L-MPZ Hom mice have ER with enlarged lumen compared with WT mice (Supplementary Fig. 8c). The mean widths of ER lumen in Schwann cells were increased in L-MPZ Hom mice compared with WT mice (Supplementary Fig. 8d). These results prove the increase of ER stress in 8-week and 10- week-old L-MPZ mouse Schwann cells. To investigate whether accumulation of L-MPZ in ER is related with ER stress in Schwann cells, we performed double immunostaining of teased sciatic nerve fibers using anti-KDEL and anti-L-MPZ

antibodies (Supplementary Fig. 8e). Confocal images with Z-stack revealed that L-MPZ signals were not colocalized with strong KDEL signals in Het and Hom L-MPZ mice Schwann cells. Therefore, increase of ER stress in L-MPZ mice Schwann cells may not be directly caused by an accumulation of excess L-MPZ in ER.

We also investigated glial fibrillary acidic protein (GFAP) as a non-myelinated Schwann cell marker and MBP as a myelin marker. L-MPZ Hom mice showed an increase of GFAP (Fig. 6a, c, and Supplementary Fig. 7; uncropped blots, Supplementary Fig. 10i, j) and decrease of MBP levels (Fig. 6a, d; uncropped blot, Supplementary Fig. 10h). Both P0 and MBP mRNA levels were reduced in L-MPZ Hom mice (Fig. 6e, f). These data indicates that specific myelin protein mRNAs including P0 mRNA producing L-MPZ were reduced according to the decrease of myelinating Schwann cells in L-MPZ Hom mice.

**Macrophage infiltration in sciatic nerves in L-MPZ mice.** In PNS pathology, macrophages are involved in neuroinflammation induced peripheral neuropathy[39]. Therefore, the number of macrophage in sciatic nerves of L-MPZ mice were examined by immunostaining with an antibody against ionized calcium binding adapter molecule 1 (Iba1), a major macrophage marker. We

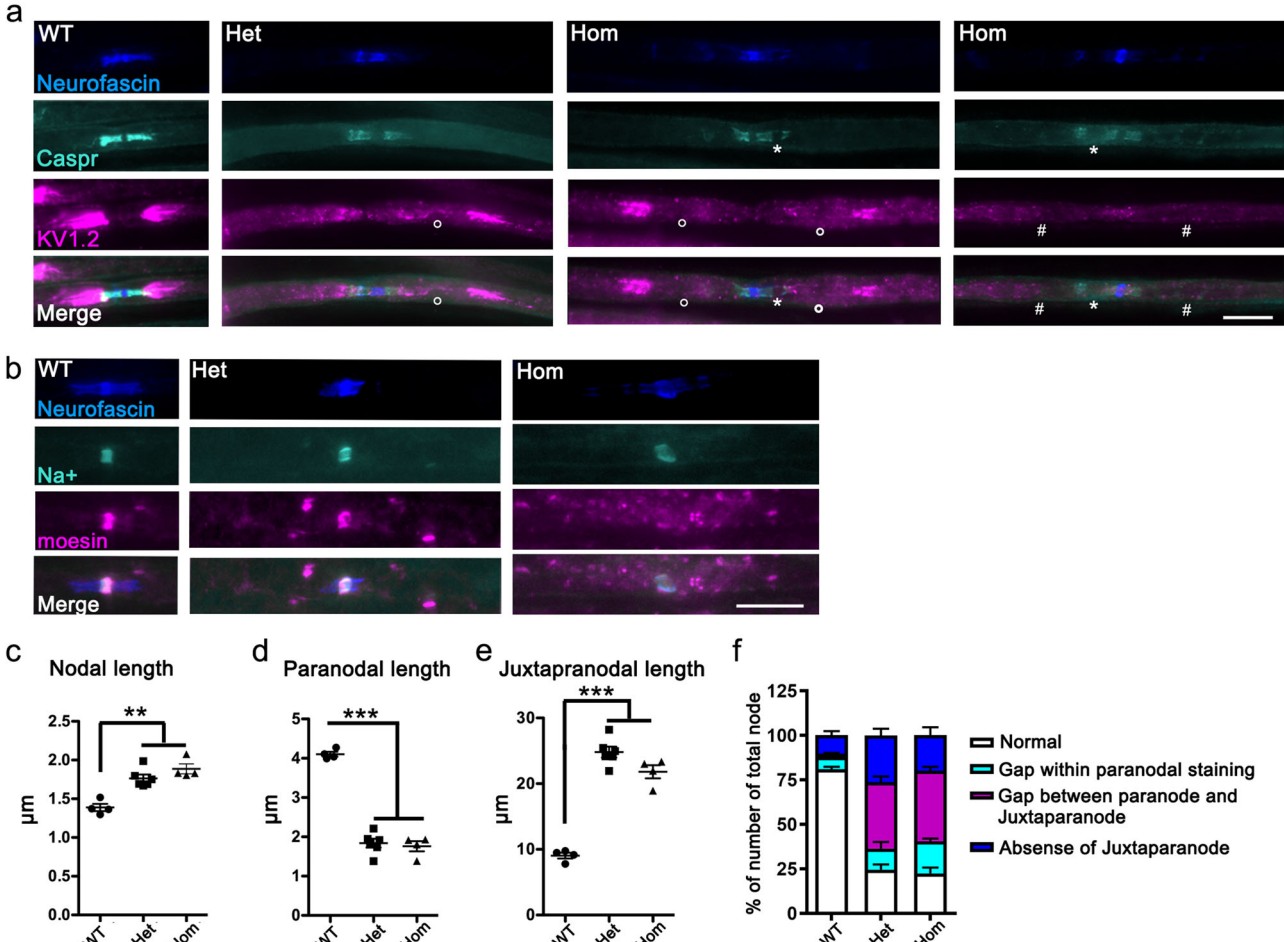

**Fig. 5 Morphological abnormalities in the node of Ranvier and surrounding regions in L-MPZ mice. a** Triple immunostaining of teased sciatic nerves in 10-week-old L-MPZ wild-type (WT), heterozygous (Het), and homozygous (Hom) mice using anti-neurofascin (blue), anti-Caspr (turquoise), and anti-Kv1.2 (magenta) antibodies. White circle (○), white asterisk (*), and # indicate either gaps between the paranode and juxtaparanode, gaps within paranodal staining, or absence of Kv1.2 clusters in the juxtaparanode, respectively. Bar, 10 μm. **b** Triple immunostaining of teased sciatic nerves using anti-neurofascin (blue), anti-sodium channel (turquoise), and anti-moesin (magenta) antibodies. Bar, 10 μm. **c–e** Measurement of nodal (sodium channel cluster; **c**), paranodal (Caspr cluster; **d**), and juxtaparanodal (Kv1.2 cluster; **e**) lengths in the sciatic nerve of L-MPZ mice. Nodal and juxtaparanodal lengths were increased while paranodal lengths were decreased in L-MPZ Hom and Het mice. **f** Quantification of the number of abnormal node–paranode–juxtaparanode structures in sciatic nerve sections of L-MPZ mice. Abnormal structures were categorized into three (turquoise column, white asterisk; magenta column, white circles; blue column, white #) as described in **a**. The percentage of nodes with abnormal node–paranode–juxtaparanode structures was increased in L-MPZ mice. **p < 0.01; ***p < 0.001 by one-way ANOVA with post-hoc Tukey's test. Data are presented as mean ± SE of experiments. N (mice) in WT = 4, Het = 6, Hom = 4 (**c–f**).

detected many strongly Iba1-positive macrophages in sciatic nerves of L-MPZ Hom and Het mice (Fig. 7a, b and Supplementary Table 1). There are several types of macrophages in healthy and disease conditions[40]. Most of these increased macrophages in 10-week-old L-MPZ Hom mice were CD68[+]/CD206[+], indicating anti-inflammatory M2a-like phenotype (Supplementary Fig. 9). Representative three-dimensional reconstructed nerve masses obtained by serial EM images using SBF-SEM are shown (Fig. 7c, WT mice, and d, L-MPZ Hom mice). Macrophages (Fig. 7e, f, colored) were more frequently observed in L-MPZ Hom mice (Fig. 7g and Supplementary Table 2). Large whorls of myelin debris were sequestered in macrophages (Fig. 7h, arrowheads) and Schwann cell bodies (Fig. 7i, arrowhead) of L-MPZ Hom mice. Thus, a decrease of myelin-specific protein levels (Fig. 6a, d) and increase of macrophages (Fig. 7) suggests demyelination in L-MPZ Hom mice.

**Increase of phosphorylation signal in L-MPZ mice.** L-MPZ has an additional PKC-mediated phosphorylation site in the L-MPZ-specific region[13]. Accordingly, PKC-phosphorylation state of L-

MPZ may be related to its unique function in the L-MPZ mouse phenotype. Hence, we determined PKC-mediated phosphorylation levels using Phos-tag western blotting. Anti-L-MPZ antibody reacted with multiple phosphorylation forms of L-MPZ in blots transferred from Phos-tag gels (Fig. 8a). Two major bands were recognized by an antibody against phospho-(Ser) PKC substrate (i and ii in Fig. 8a), suggesting that one or two sites in L-MPZ were phosphorylated by PKC in sciatic nerve. Compared with WT mice, the amount of phosphorylated L-MPZ was markedly increased in L-MPZ Hom and Het mice (Fig. 8a–c). These results suggest that an increase of PKC-phosphorylated L-MPZ may influence the L-MPZ mouse phenotype.

**Discussion**
In the present study, we investigated the function of L-MPZ using mice that synthesized L-MPZ but not P0 in PNS myelin. L-MPZ mice showed CMT-like pathologies such as myelin and axonal abnormalities and motor disturbances. Thin compact myelin with high periodicity that occasionally showed loose and abnormal

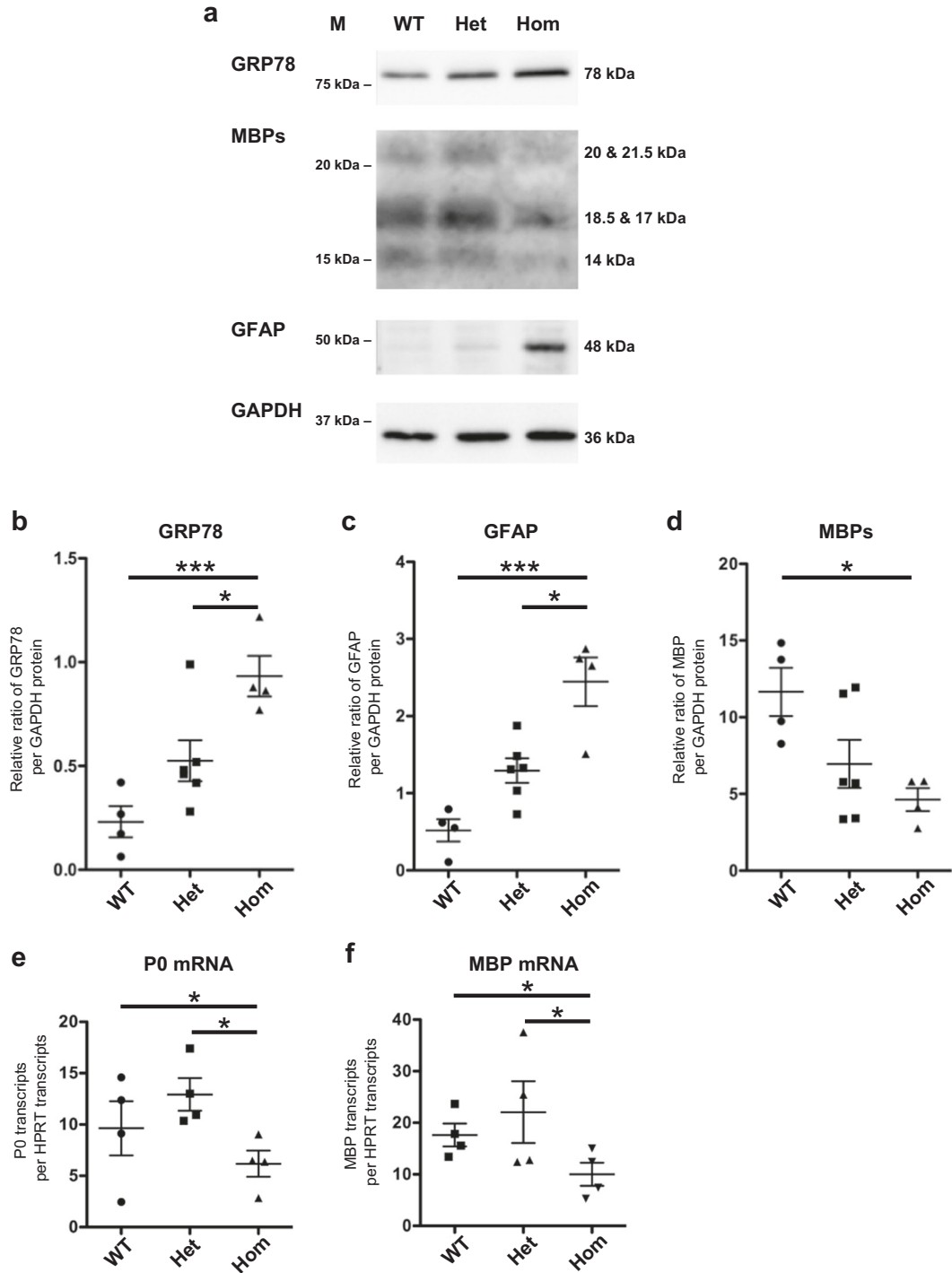

**Fig. 6 Elevation of an ER stress marker and decrease of myelin marker in L-MPZ mice. a** Immunoblotting of sciatic nerve homogenates (5 μg per lane) in 10-week-old L-MPZ mice using an ER stress marker (anti-GRP78), non-myelinated Schwann cell marker (anti-GFAP), and myelin marker (anti-MBP) antibodies. GAPDH was used as a loading control. M molecular weight marker. **b–d** Quantification of band intensities refer to **a**. L-MPZ homozygous (Hom) mice showed an increased level of GRP78 (**b**) and GFAP (**c**) levels and decrease of MBP (**d**) in the sciatic nerve. **e**, **f** qPCR analysis of sciatic nerves in L-MPZ mice. Levels of P0 (**e**) and MBP (**f**) mRNA were normalized to HPRT mRNA levels. mRNA levels of MBP and P0 were decreased in L-MPZ Hom mice. *$p < 0.05$; ***$p < 0.001$ by ANOVA with post-hoc Tukey's test. Data are presented as mean ± SE of four experiments. N (mice) in WT = 4, Het = 6, Hom = 4 (**b–d**); and WT = 4, Het = 4, Hom = 4 (**e**, **f**). Uncropped blots for **a** can be found in Supplementary Fig. 10.

protrusions characterized by wide-open MDLs, loss of cytoplasmic channel structures (including E-cadherin-positive SLI and Cajal bands), short internodal distances, increase of myelinated axons with smaller diameters, and disorganization of paranodal and juxtaparanodal structures were observed in L-MPZ mice. Increased Schwann cells containing single axons without

myelin and extensive infiltration of macrophages with phagocytosis suggests that the abnormal myelin morphology likely caused demyelination. These changes were mainly observed in Hom mice, with Het mice showing milder changes. All of these abnormalities can be responsible for the motor symptoms. Two independent L-MPZ mouse lines generated by the CRISPR–Cas9 system

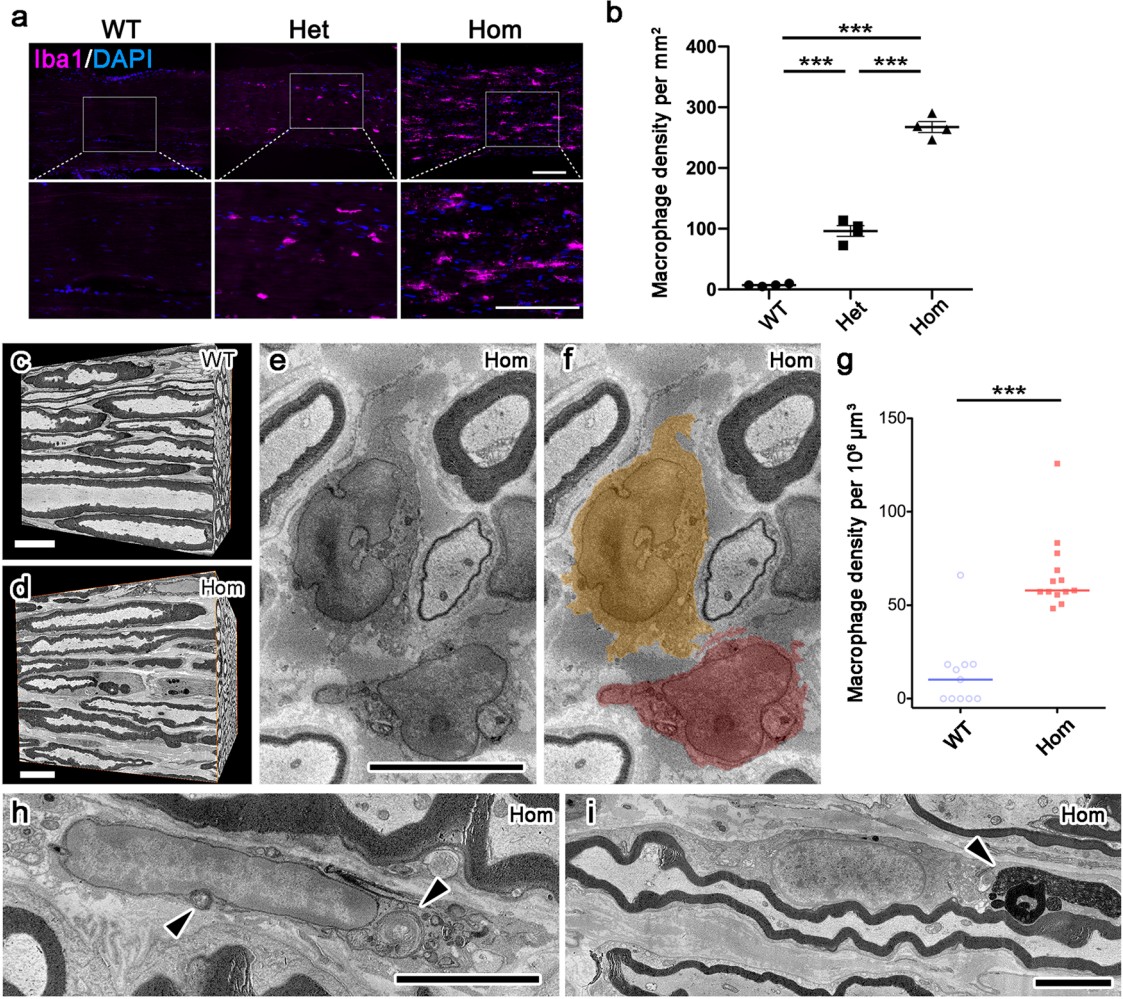

**Fig. 7 Infiltration of macrophages in L-MPZ mouse sciatic nerve. a** Immunostaining of macrophages using anti-Iba1 antibody with DAPI staining in sciatic nerve of 10-week-old L-MPZ mice. The white square is enlarged. **b** For quantification, the number of Iba1-positive macrophages per mm² was represented in each genotype. Macrophages were increased in sciatic nerves of L-MPZ homozygous (Hom) and heterozygous (Het) mice, although the number was much higher in Hom compared with Het mice. **c–i** Increased macrophage density in 8-week-old L-MPZ Hom mice by SBF-SEM. In three-dimensionally reconstructed tissue volumes obtained from serial electron microscopic images of wild-type (WT; **c**) and Hom (**d**) mice, macrophages (**e**, **f** colored) were more frequently observed in Hom mice (**g**). Large whorls of myelin debris were sequestered in macrophages (**h** arrowheads) and Schwann cells (**i**, arrowhead) of Hom mice. Bars, 100 μm (**a**), 10 μm (**c**, **d**), or 5 μm (**e**, **f**, **h**, **i**). ***$p < 0.001$ by Mann–Whitney $U$-test. Data are presented as mean ± SD of experiments. $N$ (mice) in WT = 4, Het = 4, Hom = 4 (**b**). $N$ (photo areas prepared from each of 4 mice) in WT = 11, Hom = 13 (**g**).

exhibited the same neuropathy-like phenotype indicating that these pathologies were caused by replacement of P0 with L-MPZ and not by off-target genome editing. Thus, to the best of our knowledge, this is the first report showing that disturbance of physiological translational readthrough causes a disease in mammals.

Was the phenotype of L-MPZ mice caused by deficiency of P0 protein or increase of L-MPZ in myelin? P0-null mice express severe symptoms similar to Dejerine–Sottas syndrome in humans, including an earlier onset of motor disturbance, severe hypomyelination, and onion bulb formation[10–12]. As shown here, although P0 was completely deficient (Hom) or clearly reduced (Het), the symptoms of motor disturbance in both L-MPZ Hom and Het mice were milder compared with P0-null mice. In P0-null mice, separation of extracellular IPL was observed[10]. However, in L-MPZ mice, IPLs in the remaining myelin looked normal and comparable to WT. Consequently, functional loss of P0 in extracellular IPL of L-MPZ mice appears to be compensated for by L-MPZ. Meanwhile, in L-MPZ mice, many intracellular MDLs were wide-open in myelin, and myelin outfoldings or

tomacula (which are not observed in P0-null mice[10–12]) were frequently observed (Fig. 4 and Supplementary Fig. 4a, b). The molecular difference between P0 and L-MPZ is restricted to the L-MPZ-specific domain of the cytoplasmic C-terminus. Physicochemically, the L-MPZ-specific domain is highly basic (pI ~10) and its size does not seem large enough (MW ~6.7 kDa) to separate MDLs supported by MBPs (pI > 11, MW 14–21.5 kDa). Nonetheless, as L-MPZ mice contain a large amount of phosphorylated L-MPZ proteins (Fig. 8), excessive phosphorylated polypeptides with negative charges at the cytoplasmic side of each myelin layer may work repulsively at the plasma membrane surface and/or opposed phosphorylated sites as each other, causing loose myelin, outfoldings, and higher periodicity (Fig. 4). Therefore, gain-of-excessive L-MPZ in L-MPZ mice may cause CMT-like neuropathy.

Several neuropathy model mice exhibits tomacula that are not caused by the *Mpz* gene but by other genes. Heterozygotes of peripheral myelin protein 22 (PMP22) knockout mice[2], periaxin-null mice[3,41], or O-GlcNAc transferase-conditional knockout mice[5] exhibit tomacula and a loose myelin structure, as well as

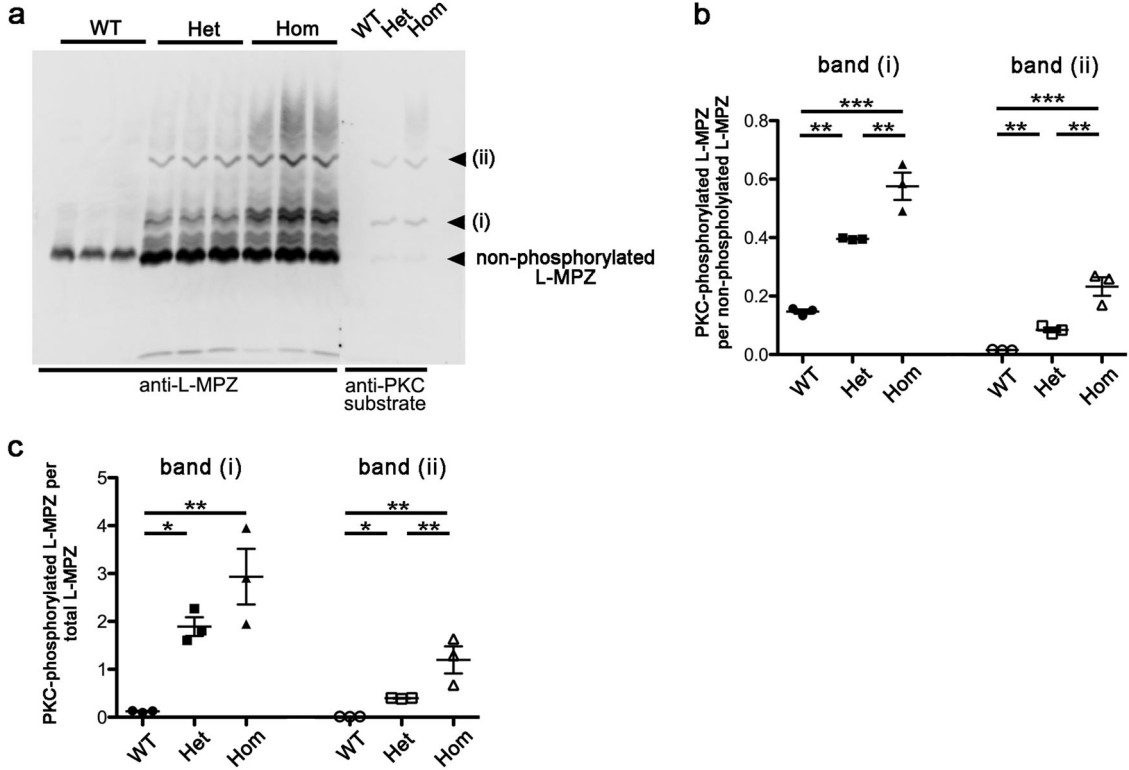

**Fig. 8 Excessive phosphorylation in L-MPZ mice. a** Phos-tag-western blotting of sciatic nerve homogenates (WT, 5 μg; Het and Hom 1 μg per lane) using anti-L-MPZ and anti-phospho-(Ser) PKC (PKC substrate) antibodies. Multiple phosphorylated-L-MPZ bands were found in both 10-week-old L-MPZ homozygous (Hom) and heterozygous (Het) mice compared with wild-type (WT) mice, in which only faintly stained bands were found. Anti-phospho-(Ser) PKC antibody recognized two L-MPZ-positive major bands (i, ii), suggesting that either a single (i) or two PKC-phosphorylation sites (ii) in L-MPZ were phosphorylated by PKC in sciatic nerves of L-MPZ Hom and Het mice. Higher smeared bands appear to be multiple-phosphorylated L-MPZ at other phosphorylation sites in addition to PKC phosphorylation. Three individual mouse samples in each group were used for anti-L-MPZ. The three lanes of anti-PKC substrate indicate representative samples in each group. **b** Each band intensity in a (anti-L-MPZ antibody) was measured and the ratio of phosphorylated L-MPZ bands (i, ii) to non-phosphorylated L-MPZ (bottom bands) were compared. L-MPZ mice showed an increased ratio of phosphorylated L-MPZ levels of these two bands especially in the sciatic nerve of Hom mice. **c** The ratio in amount of two phosphorylated L-MPZ bands (i, ii) per total L-MPZ-positive protein content was calculated and compared in different genotypes. Phosphorylated L-MPZ levels were increased in L-MPZ mice. *$p < 0.05$; **$p < 0.01$; ***$p < 0.001$ by ANOVA post-hoc with Tukey's test. Data are presented as mean ± SE of three experiments. $N$ (mice) in WT = 3, Het = 3, Hom = 3 (**b**, **c**).

similar symptoms as L-MPZ mice. Especially in the periaxin-related latter two mouse lines, loss of Cajal band structure has been shown to slow conduction velocity and impair Schwann cell elongation during nerve growth, which causes shorter internodes[3,5,41]. Thus, abnormalities in Cajal bands may cause CMT-like neuropathy in mice. Our study shows that increase of L-MPZ in myelin causes loss of Cajal band structure with an increase in number of myelinated fibers with shorter internodal lengths in L-MPZ mice (Fig. 2e, f, and Supplementary Fig. 5). Altogether, these results indicate that disorganization of Cajal bands may be partly responsible for the neuropathy phenotype in L-MPZ mice.

In many neurodegenerative diseases including Alzheimer's disease, Parkinson's disease, polyglutamine disease, and amyotrophic lateral sclerosis, involvement of ER stress has been suggested[42]. In myelinating cells, ER stress is also a possible cause of demyelination[43]. For example, the CMT1B mouse model (P0S63del mutant mouse) has shown that mutant P0 is retained in the ER, and that the protein kinase R (PKR)-like endoplasmic reticulum kinase (PERK) pathway is activated to increase the phosphorylated α subunit of eukaryotic initiation factor 2 (eIF2α), activating transcription factor 4 (ATF4), and C/EBP homologous protein (CHOP) in Schwann cells[37,44]. As shown in Fig. 6a, b, and Supplementary Figs. 7, 8a–d, signs of ER stress were clearly

increased in Schwann cells in L-MPZ mice. Therefore, ER stress induced by excessive L-MPZ is probably involved in development of CMT-like pathology. However, since in L-MPZ mouse Schwann cells, excessive L-MPZ molecules were apparently not accumulated in ER (Supplementary Fig. 8e), ER stress may be induced by a different indirect mechanism from that existing in previous P0 mutant mice. Further detailed analyses are required to clarify the induction mechanism of ER stress in L-MPZ mice.

Lower CMAP amplitude in addition to slower NCV (Fig. 1) indicates the presence of not only myelin but also axonal damage in L-MPZ nerve. Indeed, morphological studies confirmed these changes. The most prominent axonal abnormalities were an increase of small-caliber axons (Fig. 3b, h) and disorganization of functional axonal domains represented by specific markers, which were further confirmed by EM analyses (Fig. 5a–f and Supplementary Fig. 6). It has previously been reported that ablation of paranodal components leads to loss of paranodal axo–glial junctions and disorganization of axonal domain molecule distribution in myelinated peripheral nerves, with a resulting decrease in NCV and CMAP amplitudes[29,31]. Thus, in L-MPZ mice, discontinuous paranodal formation and loss of paranodal axo–glial junctions in some myelinated fibers may cause electrophysiological abnormalities. Furthermore, in the extracellular matrix surrounding nodes, which acts on stabilization of nodal

proteins by axonal cytoskeletal scaffolds[45,46], Schwann cell microvilli were disorganized and relative diffusion of sodium channels was observed (Fig. 5b). Although the mechanism that affects axonal diameter is still uncertain, weak adhesion property at MDLs and loss of cytoplasmic channels in L-MPZ mouse myelin may lead to dysfunction of axo–glia interactions.

L-MPZ Het mice only produce P0 from a normal allele of the *Mpz* gene. The protein ratio of P0 and L-MPZ in peripheral nerve of Het mice was around 1:1, with their resulting CMT-like neuropathy symptoms being milder than those of L-MPZ Hom mice. In contrast, P0-null Het mice showed normal development and maintenance of myelin until 4-months-old, but later displayed CMT-like neuropathy[11]. The onset of the neuropathy phenotype became apparent at 10 weeks of age in L-MPZ Het mice (Fig. 1), which is earlier than in P0-null Het mice[12], therefore increased L-MPZ/P0 ratio, rather than decreased amount of P0, may be related to phenotype severity. This indicates that an appropriate L-MPZ/P0 ratio (translational readthrough ratio) is required for formation and/or maintenance of myelin morphology and proper function.

In summary, we have shown that changes in amount of L-MPZ influences the formation and maintenance of myelin. Therefore, analyses of gene mutations that influence L-MPZ production and its function may be required for CMT patients of unknown etiology. In addition, many readthrough agents have recently been developed for treatment of severe hereditary diseases caused by nonsense mutations such as Duchenne muscular dystrophy[47–50]. Because an increase of L-MPZ/P0 ratio induces CMT-like neuropathy in mice, and L-MPZ is also produced by the same mechanism in humans, we should consider possible side effects when readthrough agents are used for treatment of diseases. Together with the present study, future studies that focus on changes of peripheral nerve in the absence of L-MPZ may ascertain the functional significance of the physiological readthrough mechanism in the PNS. Thus, L-MPZ mice will become a unique model of neuropathy caused by alteration of translational readthrough.

## Methods

**Materials**. Chemicals were of the highest purity and obtained from Fujifilm Wako Pure Chemical Co., (Osaka, Japan), Sigma–Aldrich Japan/Merck (Tokyo, Japan), Nacalai Tesque (Kyoto, Japan), and Kanto Chemical (Tokyo, Japan).

**Mice**. L-MPZ mice were generated by replacement of the canonical stop codon (TAG) with an alanine codon (GCG) using the CRISPR–Cas9 genome editing system[51]. L-MPZ founder mice were provided from the Laboratory Animal Resource Center, University of Tsukuba. For backcrossing, C57BL/6J mice were purchased from Charles River Laboratories Japan (Yokohama, Japan).

Two founder mouse lines that transmitted the replaced genes to their offspring were produced and confirmed by genomic sequencing. L-MPZ mice were maintained on a C57BL/6J background. Genotypes were determined by allele-specific PCR, with primer design as described in Hayashi et al.[52]. Allele-specific PCR of mouse tail genomic DNA was performed using Go Taq Green Master Mix (Promega, Madison, WI, USA) with the following S set primers: P0 STOP Allele Forward (P0_Stop_Allele_Fw), 5′-GGAGTCTCGCAAGGATAAGAGAT-3′ and MPZ mutant genome reverse (MPZ_Mut_Geno_Rv), 5′-TGGTTCTACTGGGGGACTTGACTG-3′; and A set primers: P0_MutA_Allele_Fw, 5′-GGAGTCTCGCAAGGATAAGAGAG-3′ and MPZ_Mut_Geno_Rv. The S primer set generated a 167 bp PCR product from WT and Het genomic DNA, while the A primer set generated a 167 bp product from Het and Hom genomic DNA (Supplementary Fig. 1b). PCR conditions were as follows: pre-denaturation at 95 °C for 2 min, followed by 32 amplification cycles of denaturation at 95 °C for 30 s, primer annealing at 62 °C for 30 s, and extension at 72 °C for 30 s, with a final additional extension at 72 °C for 5 min. For genotyping at protein level, each mouse nerve homogenates were analyzed on SDS-PAGE gels stained with coomassie brilliant blue R-250, or western blots. Heterozygous animals were used for colony maintenance. Littermates or age-matched WT mice were used as control animals. Mouse lines were maintained in a designated specific-pathogen-free environment at the animal facility of Tokyo University of Pharmacy and Life Sciences under university guidelines for the care and use of animals. All animal protocols were approved by the Institutional Animal Use Committee at Tokyo University

of Pharmacy and Life Sciences (approval number: P16-95, P17-68, P18-76, and P19-46).

**Homogenization**. Sciatic nerve homogenates were prepared from 10-week-old L-MPZ WT, Het, and Hom mice. Mice were anesthetized by intraperitoneal injection of anesthetic combination comprising 0.3 mg kg$^{-1}$ medetomidine, 4.0 mg kg$^{-1}$ midazolam, and 5.0 mg kg$^{-1}$ butorphanol[53]. All procedures were performed on ice or at 4 °C. Homogenates were obtained as described previously[13,14,54] with minor modifications. In brief, sciatic nerves were dissected and snap-frozen in liquid nitrogen. Frozen tissue was then ground into a powder using BioMasher II (Nippi, Tokyo, Japan) in 0.32 M sucrose containing 5 mM Tris-HCl (pH 7.5), 2 mM ethylene glycol tetraacetic acid, 0.75 μM aprotinin, 1 μM leupeptin, 1 μM pepstatin A, and 0.4 mM phenylmethylsulfonyl fluoride (homogenization buffer), and if necessary, PhosSTOP (Roche Diagnostics, Mannheim, Germany) was added to prevent dephosphorylation. To remove chromosomal DNA, cell debris and fibers, homogenates were centrifuged at 800×*g* for 10 min, and supernatants were collected and stored as whole homogenate fractions at −80 °C. Protein concentration was determined using a bicinchoninic acid assay (Takara Bio, Shiga, Japan). Samples were examined by SDS–PAGE or Phos-tag SDS–PAGE and immunoblot analysis with various antibodies.

**Western blotting and Phos-tag western blotting**. Western blotting was performed as described previously[54] with minor modifications. Briefly, each sample was separated on either 7.5% or 12% SDS–PAGE gels and transferred to polyvinylidene fluoride membranes (Merck Millipore, Billerica, MA, USA). Membranes were incubated for 1 h with blocking buffer containing 0.3% skimmed milk or 2% donkey serum in 50 mM Tris-HCl (pH 7.4), 150 mM NaCl, and 0.05% Tween 20 (T-TBS) at room temperature. Membranes were then incubated for 1 h with primary antibodies diluted in blocking buffer and washed three times in T-TBS prior to incubation for 1 h with secondary antibodies in T-TBS. After washing three times in T-TBS, immunoreactivities were detected using the ECL system (GE Healthcare Japan, Tokyo, Japan). Both chemiluminescence and prestained size marker images were captured at the same field of view using LuminoGraphI (WSE-6100; ATTO corporation, Tokyo, Japan). Size markers were aligned with positive bands on the same blot using software (Photoshop CS5; Adobe systems Inc, San Jose, CA, USA). For quantification, band intensities were measured using ImageJ (https://imagej.net/Welcome). To distinguish phosphorylated proteins, Phos-tag (50 μM)-containing 7.5% SDS–PAGE was performed, followed by western blotting according to the manufacturer's instructions (Fujifilm Wako Pure Chemical Co.).

**RNA extraction and quantitative real-time PCR**. Anesthetized mice using anesthetic combination (0.3 mg kg$^{-1}$ medetomidine, 4.0 mg kg$^{-1}$ midazolam, and 5.0 mg kg$^{-1}$ butorphanol) were sacrificed by cervical dislocation. Sciatic nerves were dissected and snap-frozen in liquid nitrogen. Frozen tissue was then ground into a powder using BioMasher II. Total RNA was extracted using Trizol (Invitrogen, Carlsbad, CA, USA). Quantification of mRNA of P0, MBP, or the housekeeping gene, hypoxanthine phosphoribosyl transferase (HPRT), was performed using a fluorescence-based real-time detection method. First strand cDNA was synthesized using ReverTra Ace qPCR kit (Toyobo, Osaka, Japan). Real-time PCR was then performed using the CFX Connect Real-Time PCR Detection System (Bio-Rad Laboratories, Hercules, CA, USA) with the primer sets: P0_primer_Fw, 5′-CTTCAAGATGGACGCGACAC-3′ and P0_primer_Rv, 5′-AGAGTGTCT CAGCCTCCACAG-3′; MBP_primer_Fw, 5′-AACATTGTGACACCTCGAACA-3′ and MBP_primer_Rv, 5′-TGTCTCTTCCTCCCCAGCTA-3′; HPRT_primer_Fw, 5′-CCCTCTGGTAGATTGTCGCTTA-3′ and HPRT_primer_Rv, 5′-AGATGCTGTTACTGATAGGAAATCGA-3′. Relative copy number of P0 or MBP transcripts per HPRT transcripts was determined using calibration standards for each of the tested molecules.

**Tail suspension test and rotarod test**. To detect motor dysfunction, usually hindlimb adduction (clasping) is checked by the tail suspension test[55–57]. To quantitatively evaluate the severity of motor impairments, a modified method of the tail suspension test was used, which has been adopted for screening of potential antidepressant drugs[58]. To analyze motor function, 10-week-old L-MPZ WT, Het, and Hom mice were suspended from the tail and video recordings obtained for 6 min. To examine motor function, the angles formed between bilateral hindlimbs were measured at two time points after starting the tail suspension: 1–2 min (early period) and 4–5 min (later period) (Fig. 1a and Supplementary Movies 1–6).

Motor function was also examined using a MK-630B rotarod (Muromachi Kikai Co., Tokyo, Japan). Mice were placed on a rubber-coated rod and two test trials were performed with an inter-trial interval of 10 min. After test trials, the rotarod test was performed. Rotation speed of the rod was 20 rpm with each 5 min trial. The latency was measured when the mouse stopped walking on the rod, either by falling off or by clinging to the rod, and mean latency from three independent trials was calculated for each mouse.

**Electrophysiological analysis**. Motor NCVs were measured as previously described[59] with slight modifications. In brief, conduction speeds in motor nerve fibers innervating the plantar muscle was examined under anesthesia by

intraperitoneal injection of anesthetic combination (0.3 mg kg$^{-1}$ medetomidine, 4.0 mg kg$^{-1}$ midazolam, and 5.0 mg kg$^{-1}$ butorphanol). Body temperature was monitored at the thigh and maintained around 30 °C using a heating pad. As the anode and cathode, two stainless steel subdermal needle electrodes (0.5× 27 G; Axon Systems, Hauppauge, NY, USA) were placed in the plantar muscle for detection of CMAPs. Proximal and distal stimulation electrodes were inserted in the ankle and sciatic notch, respectively. Electrical stimulation (0.1 ms in duration) was applied every second using a constant voltage stimulator (DPS-007; Dia Medical Systems, Tokyo, Japan). To detect the amplified waveforms, a digital data recorder (NR-2000; Keyence, Osaka, Japan) was used. For recording and analyzing, WAVE SHOT 2000 software (Keyence) was used. The trigger level was set to 0.1 V for recording. Data were recorded for 10 ms (every 5 μs, 2000 samples) from 500 μs before the trigger. Distance between the distal and proximal electrodes and the difference in onset latency obtained by stimulation of each electrode were used for calculation of conduction velocity. Waveform duration was measured between the onset and endpoint of each CMAP.

**Immunofluorescence study.** Immunostaining methods were modified as described previously[59]. Sciatic nerves obtained from anesthetized 10-week-old mice were rapidly dissected and immediately fixed in ice-cold 4% paraformaldehyde (PFA) in 0.1 M phosphate buffer (PB) (pH 7.4) for 30 min. After fixation, nerves were cryoprotected with 10% sucrose in 0.1 M PB, and then later with 30% sucrose overnight at 4 °C. Next, nerves were frozen in OCT compound (Sakura Finetek, Tokyo, Japan). Sections (10 μm) were cut using a cryostat (CM1850; Leica Microsystems, Wetzlar, Germany), and placed on FRONTIER slide glass (Matsunami Glass, Osaka, Japan). For teased fiber preparation, sciatic nerves were transferred into 0.1 M PB (pH 7.4) after fixation. The nerves were then gently teased apart, spread on FRONTIER slide glass, and air-dried. If necessary, samples were incubated with −20 °C methanol for 5 min, and rinsed in phosphate buffered saline (PBS) before pre-incubation with 0.1 M PB (pH 7.4) containing 0.3% Triton X-100 and 10% normal goat serum (PBTGS) for 1 h. Afterwards, samples were incubated overnight at 4 °C with primary antibodies diluted to appropriate concentrations in PBTGS. Samples were thoroughly rinsed in PBS, followed by application of fluorescently labeled secondary antibodies (1:2000 in PBTGS) for 1 h at room temperature. Finally, labeled samples were rinsed with 0.01 M PBS every 5 min, three times. Samples were mounted on cover glasses using Vectashield with DAPI (Vector Laboratories, Burlingame, CA, USA) and examined by BZ-X700 microscopy (Keyence) or FV-1000 confocal microscopy (Olympus, Tokyo, Japan).

**Antibodies.** Rabbit polyclonal anti-L-MPZ antibody (immunostaining, 1:4000; Western blotting, 1:40,000) was produced as previously described[13]. Rabbit polyclonal antibodies against Kv1.2 (immunostaining, 1:400; Chemicon, Merck Millipore, discontinued), GFAP (immunostaining, 1:10; Western blotting, 1:1000; DAKO, Carpinteria, CA, USA), phospho-(Ser) PKC substrate (Western blotting, 1:1000; Cell Signaling, Danvers, MT, USA), DRP2 (immunostaining, 1:50; Sigma–Aldrich), GRP78 (BiP) (immunostaining, 1:200; Western blotting, 1:2000; abcam, Cambridge, UK), Iba1 (immunostaining, 1:800; Fujifilm Wako Pure Chemical Co.), and Mannose receptor (CD206) (immunostaining, 1:200; abcam), chicken polyclonal antibody against Neurofascin (immunostaining, 1:400; R&D Systems, Minneapolis, MN, USA), mouse monoclonal antibodies against Caspr (1:400; clone K65/35; NeuroMab, Davis, CA, USA), sodium channel (Pan) (immunostaining, 1:400; clone K58/35; Sigma–Aldrich), beta dystroglycan DAG$_{43}$/DAG$_{31}$ (Western blotting, 1:1000; clone 43DAG1/8D5; Leica Biosystems, Newcastle upon Tyne, UK), moesin (immunostaining, 1:100; clone M22; Sanko Junyaku, discontinued), KDEL (immunostaining, 1:100; clone 10C3; Enzo Life Sciences, Inc., Farmingdale, NY, USA), actin, (Western blotting, 1:1,0000; Fujifilm Wako Pure Chemical Co.), GAPDH (Western blotting, 1:10,000; Fujifilm Wako Pure Chemical Co.), and CD68 (immunostaining, 1:1000; clone ED1; abcam), rat monoclonal antibodies against MBP (immunostaining, 1:200; Western blotting, 1:4000; Chemicon, Merck Millipore), laminin alpha-2 (Western blotting, 1:1000; clone 4H8-2; Santa Cruz Biotechnology, Dallas, TX, USA), and E-cadherin (immunostaining, 1:100; clone ECCD-2; Takara Bio, Kusatsu, Japan), and goat polyclonal antibody against P0 (1:2000; Abnova, Taipei, Taiwan) were purchased. Secondary antibodies including Alexa 488-conjugated and 594-conjugated species-specific antibodies (1:2000; Molecular Probes/Thermo Fisher Scientific, Waltham, MA, USA), and aminomethyl coumarin-conjugated anti-chicken IgY antibody (1:500; Jackson ImmunoResearch Laboratories, West Grove, PA, USA) were used for immunohistochemistry. For western blotting, horseradish peroxidase-conjugated anti-mouse, anti-rabbit, anti-rat, and anti-goat IgG antibodies (1:10,000; Jackson ImmunoResearch Laboratories) were used.

**Electron microscopy analyses.** Eight-week-old mice were deeply anesthetized by intraperitoneal injection of anesthetic combination. Four WT and four L-MPZ Hom mice were perfused with 4% PFA and 2.5% glutaraldehyde in 0.1 M PB (pH 7.4), and incubated in the same fixative overnight. For TEM, samples were postfixed in 2% osmium tetroxide (Nisshin EM Co., Tokyo, Japan) in PBS, dehydrated in a graded ethanol series, treated with dehydrated acetone, and embedded in Quetol 812 epoxy resin (Nisshin EM Co.). For observation by SBF-SEM, sample

preparation, observation, and analyses of acquired data were performed as described previously with slight modifications[60]. Briefly, samples were treated with 2% osmium tetroxide (Nisshin EM Co.) and 1.5% potassium ferrocyanide in PBS for 1 h at 4 °C, followed by 1% thiocarbohydrazide for 20 min at room temperature, 2% aqueous osmium tetroxide for 30 min at room temperature, 2% uranyl acetate solution overnight at 4 °C, and lead aspartate solution for 30 min at 65 °C. Lead aspartate solution was produced by adding lead nitrate (TAAB, Aldermaston, UK) into 0.03 M L-aspartic acid to a final concentration of 0.67% (pH 5.0–5.5). Samples were then dehydrated in a graded ethanol series, treated with dehydrated acetone, and embedded in Quetol 812 epoxy resin containing Ketjen black powder[61]. The resin was incubated at 70 °C for three nights to ensure polymerization. SBF-SEM observation of the myelin membrane was performed using Sigma or Merlin scanning electron microscopes (Carl Zeiss, Oberkochen, Germany) equipped with a 3View in-chamber ultramicrotome system (Gatan, Pleasanton, CA, USA). Serial image sequences were 6144 × 6144 pixels wide (4.7–6 nm per pixel) and 200–1000 slices in 80 nm steps. Sequential images were processed using FIJI (https://fiji.sc/), and analyses were performed using Amira (FEI, Tokyo, Japan).

For the percentage of axons without compact myelin that were individually ensheathed by Schwann cells, axons were counted in single EM images. Remak bundles were excluded from analyses. Diameters of axons and myelinated fibers were obtained by dividing the perimeter with π. Periodicity measurement was performed with FIJI by obtaining mean values of periodicity, and medians of outer, middle, and inner parts (three measurements in each part) in individual myelinated fibers. Macrophages in SBF-SEM images were identified as cells with relatively short but complex processes with a ruffled surface and abundant vacuoles and primary and secondary lysosomes. The number of macrophages was divided by the observation volume to obtain a macrophage density (per $10^6$ μm$^3$).

The width of each ER profile was measured as the largest distance of the membranes which was perpendicular to the longer axis of the profile. The mean ER width of each cell was pooled from three wild-type and three homozygous mice.

**Hematoxylin and eosin staining.** Muscle tissues were harvested from 4-month-old L-MPZ WT, Het, and Hom mice, and flash-frozen using dry ice–isopentane. Sections (10 μm) were cut by cryostat and placed on FRONTIER slide glass. For histological staining, slides were incubated with Mayer's hematoxylin solution (Fujifilm Wako Pure Chemical Co.) for 15 min, followed by a 10 min wash. Slides were then incubated in 1% eosin solution (Fujifilm Wako Pure Chemical Co.) for 1 min. Next, slides were washed with tap water for 10 min. Afterwards, they were dehydrated successively with 70, 80, and 90% ethanol (EtOH) for 1 min, three times with 100% EtOH for 1 min, and three times with xylene for 1 min. Samples were mounted with MGK-S (Matsunami Glass) and covered by cover slips, and observed using a BZ-X700 microscope (Keyence).

**Statistics and reproducibility.** Statistical analyses were performed using PRISM 5 (GraphPad Software, La Jolla, CA, USA). Data were expressed as mean ± standard error (SE) or standard deviation (SD). For comparisons, Mann–Whitney $U$-test, Student's $t$-test, or one-way analysis of variance (ANOVA) followed by Tukey's post-hoc test or two-way ANOVA followed by Bonferroni multiple comparison tests were performed. $p$-values < 0.05 were regarded as statistically significant. Details of each statistical test are indicated in figure legends. Non-numeric data are shown as representative results from more than three independent experiments.

**Reporting summary.** Further information on research design is available in the Nature Research Reporting Summary linked to this article.

## Data availability

All data generated during and/or analyzed during this study are included in this published article and its supplementary information. The source data underlying plots are presented in Supplementary Data 1. Full blots are shown in Supplementary Information. Any detailed data supporting the findings of this study are available from the corresponding author upon reasonable request.

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

## Acknowledgements

The authors thank D. Yamaura, M. Takehara, T. Nakajima, and T. Narazaki at Tokyo University of Pharmacy and Life Sciences, H.B. Nguyen at National Institute for Physiological Sciences, and M. Yatabe at Jichi Medical University for their technical assistance. The authors also thank Dr. S. Takahashi and Dr. S. Mizuno for generating L-MPZ founder mice. This work was supported by the Japan Society for the Promotion of Science (JSPS) KAKENHI (Grant Number 24500449), JSPS KAKENHI (Grant Number JP 16K07069), JSPS KAKENHI (Grant Number JP 16H06280), JSPS KAKENHI (Grant Number JP 19K06889), JSPS KAKENHI (Grant Number JP 19K16267), the Sasakawa Scientific Research Grant from the Japan Science Society (Grant Number 2019–4061), Grant-in-Aid for Scientific Research on Innovative Areas-Resource and technical support platforms for promoting research "Advanced Bioimaging Support", and Ministry of Education, Culture, Sports, Science and Technology (MEXT)-Supported Program for the Strategic Research Foundation at Private Universities, 2015–2019. We thank Rachel James, Ph.D. for editing a draft of this manuscript.

## Author contributions

Y.O., Y.Y., and H.B. designed the research and wrote the manuscript; Y.O. performed all experiments and data analyses except EM analyses; Y.Y. established and maintained L-MPZ mice lines; J.C. performed immunohistochemical analyses; and N.O. performed EM analyses.

## Competing interests

The authors declare no competing interests.
