## [Peer Review File · Communications Biology]

Editorial Note: *This manuscript has been previously reviewed at another Nature Research journal. This document only contains reviewer comments and rebuttal letters for versions considered at Communications Biology.*

Reviewers' comments:

Reviewer #3 (Remarks to the Author):

In recent work, the authors of this manuscript identified L-MPZ as an isoform of MPZ generated via translation read-through at the C-terminus of the protein.

To understand the function of peptides naturally generated via this mechanism Otani and colleagues generated mice harboring MPZ alleles genetically modified to express L-MPZ rather than MPZ. They convincingly show that the increase L-MPZ/MPZ ratio (normally L-MPZ represent 10% or less of MPZ) determines a dose dependent CMT-like phenotype. Both Het and Homo L-MPZ mice develop behavioural, neurophysiological and morphological deficits typical of hereditary neuropathies, although the defects are more evident in the homozygous state.

The work is technically sound, statistical methods seem appropriate and the overall results appear convincing in claiming that increased L-MPZ leads to a neuropathy.

However there are still some issues that need some addressing:

1) The authors suggest that activation of ER-stress could be part of the pathomechanism. Although it has been shown that ER-stress and the activation of the Unfolded Protein Response (UPR) are one of the main disease mechanism in CMT1B (Wrabetz et al. 2006; Saporta et al., 2012; Pennuto et al., 2008; Bai et al., 2018), this is normally caused by ER-retention of mutant MPZ proteins. Here the authors show increase in BiP and claim also increase in P-PERK (more on this below). These markers are no doubt sensors of ER-stress. However they still fails to show any direct link between L-MPZ expression and stress. In other words the authors should test if L-MPZ is ER-retained. This is in reality not so difficult, and it has been done multiple time by other groups (See for example Pennuto et al., 2008; Fratta et al. 2011). An immunofluorescence for L-MPZ (on the homozygous mice), on teased fibers or transverse sections, coupled to staining for ER-markers such as Grp78, KDEL, or calnexin, should reveal if the L-MPZ protein is partially retained. The authors in fact show somehow contradictory results: they claim ER-stress due to L-MPZ but then show colocalization in myelin with MBP. So the question of whether it is L-MPZ to activate stress remains open and unanswered. Should the immunofluorescence not be conclusive, the authors could trasfect L-MPZ and chek if, as compared to normal MPZ it is more retained, and if it alters cell adhesion.

2) Related to the previous point, the authors claim to show PERK in figure S7A. However the figure appear to show CD206, just like figure S8A (they are exactly the same images). There is no PERK data reported in this manuscript. Also, generally it is believed that anti-P-PERK antibodies do not work very well. Maybe a WB for P-eIF2alpha would be more informative.

Minor concerns:

3) Figure 2C: the SLI shown in figure 2 are far too small. Bigger magnification is necessary.

4) It would be good to see a WB for periaxin, to see how this relates to the abnormalities in Cajal bands.

5) One more point related to ER-stress: the enlarged ER should be somehow quantified. How many SC

show this feature in L-MPZ mice?

The followings are our responses to Reviewer #3's comments:

We greatly appreciate again your valuable comments and suggestions.

Comment 1: *The authors suggest that activation of ER-stress could be part of the pathomechanism. Although it has been shown that ER-stress and the activation of the Unfolded Protein Response (UPR) are one of the main disease mechanism in CMT1B (Wrabetz et al. 2006; Saporta et al., 2012; Pennuto et al., 2008, Bai et al., 2018), this is normally caused by ER-retention of mutant MPZ proteins. Here the authors show increase in BiP and claim also increase in P-PERK (more on this below). These markers are no doubt sensors of ER-stress. However they still fails to show any direct link between L-MPZ expression and stress. In other words the authors should test if L-MPZ is ER-retained. This is in reality not so difficult, and it has been done multiple time by other groups (See for example Pennuto et al., 2008; Fratta et al. 2011). An immunofluorescence for L-MPZ (on the homozygous mice), on teased fibers or transverse sections, coupled to staining for ER-markers such as Grp78, KDEL, or calnexin, should reveal if the L-MPZ protein is partially retained. The authors in fact show somehow contradictory results: they claim ER-stress due to L-MPZ but then show colocalization in myelin with MBP. So the question of whether it is L-MPZ to activate stress remains open and unanswered. Should the immunofluorescence not be conclusive, the authors could trasfect L-MPZ and chek if, as compared to normal MPZ it is more retained, and if it alters cell adhesion.*

Response 1: To support the result of Grp78 analysis, we demonstrated that an additional ER marker, KDEL signal intensity per area was significantly increased in L-MPZ Hom mouse sciatic nerves compared with WT. Additionally, according to your suggestion, we performed double immunostaininag of teased sciatic nerve fibers of 10-week-old L-MPZ mice using anti-KDEL and anti-L-MPZ antibodies and carefully examined the confocal images with Z-stack. This experiment revealed that L-MPZ signals were apparently not colocalized with KDEL signals in Het and Hom L-MPZ mice Schwann cells, suggesting ER stress may not be directly induced by an accumulation of excess L-MPZ in ER. We have added these data as Supplementary Figure S8a, b and e, and described them in “Results (page 12, line 23– page 13, line 14) and Discussion (page 17,

“ER stress...” section, line 12–19). Based on these results, at present, we did not perform cell culture analysis.

Comment 2: *Related to the previous point, the authors claim to show PERK in figure S7A. However the figure appear to show CD206, just like figure S8A (they are exactly the same images). There is no PERK data reported in this manuscript. Also, generally it is believed that anti-P-PERK antibodies do not work very well. Maybe a WB for P-eIF2alpha would be more informative.*

Response 2: First, we apologize the careless mistake in Supplementary Figure S7 in previous submission. In the previous manuscript, we decided to exhibit the image of the obvious increase of anti-p-PERK positive signals in L-MPZ Hom mice sciatic nerve sections. However, since, as you pointed out, there is some concern about the specificity of used anti-p-PERK antibody (CUSABIO), we omitted the p-PERK data from this revised manuscript. In addition, according to your suggestion, we examined western blotting using newly purchased anti-eIF2alpha and anti-p-eIF2alpha antibodies (Cell Signaling Technology, #9722 and #3398). Nevertheless, especially since titer of anti-p-eIF2alpha was very weak, we could not get reliable result.

At present, due to the problem of specific antibody availabilities, we have not clarified the onset mechanism of ER stress in L-MPZ mouse Schwann cell. Since, as shown above, ER stress seems to be secondary change caused by abnormal increase of L-MPZ and a main issue of this manuscript is that increase of L-MPZ causes CMT-like pathology, we suppose that the research to clarify a detailed correlation of increase of L-MPZ with ER stress in Schwann cell is a next step.

Minor concerns

Comment 3: *Figure 2C: the SLI shown in figure 2 are far too small. Bigger magnification is necessary.*

Response 3: As you pointed out, images of Figure 2c are too small. Therefore, we took Figure 2e from the previous version, as Supplementary Figure S3, and enlarged Figure 2c.

Comment 4: *It would be good to see a WB for periaxin, to see how this relates to the abnormalities in Cajal bands.*

Response 4: We have shown the data of western blotting using anti-laminin alpha-2 and beta-dystroglycan, which supported the decrease of DRP2-positive area indicating loss of Cajal bands. At present, since these results are enough to demonstrate the disruption of Cajal bands, we think that the data of western blot of periaxin will not be necessary for this manuscript. The detailed mechanism losing of Cajal band caused by the increase of L-MPZ will be clarified as a next stage including an analysis of periaxin signaling pathway.

Comment 5: *One more point related to ER-stress: the enlarged ER should be somehow quantified. How many SC show this feature in L-MPZ mice?*

Response 5: According to your comment, we quantified widths of ER lumen of Schwann cells in WT and L-MPZ Hom mice and added a graph as Supplementary Figure S8e. This measurement was described in “Methods” (P-27, line 20–22).

Additional changes:

- According to changes described above, we changed numbering of Supplementary Figures (S3 – S9) and related text regions.
- We added additional description about magnification of images in Supplementary Figure S2 legend.

Minor corrections:

- P-3 line 8: regions of Schwann cells → regions of myelinated Schwann cells
- P-7 line 2: Fig. 1a and b, WT; → Fig. 1a and b;
- P-13 line 15: next → also
- P-25 “Antibodies” section:
 - removed Phospho-EIF2AK3 antibody and added KDEL antibody
- Supplementary Fig. S1b and c: added Markers

REVIEWERS' COMMENTS:

Reviewer #3 (Remarks to the Author):

In this revised manuscript the authors have properly answered to the reviewer's comments. As a result, the manuscript has been improved.

No further comments